# Clothing-disentangled 3D character generation from a single image

## Abstract

This paper tackles the challenge of generating clothing-disentangled 3D characters from a single image. Existing approaches typically employ multi-layer 3D representations to model the body and each garment and then iteratively optimize these representations to fit the observations, which is time-consuming and not scalable. To address this, we propose the first feed-forward method enabling efficient and robust clothing disentanglement. Our approach first generates the multi-view images for each component of the clothed character and then employs a generalizable multi-view reconstruction method to create the 3D models of each component. For high-quality disentanglement, we propose a two-stage disentanglement approach that first disentangles each component in the 2D image space and then generates the multi-view images for each part. During the 2D component disentanglement stage, we introduce a novel multi-part diffusion model that allows information exchange among different components. Additionally, for component combination, we incorporate a novel combination attention mechanism into the multi-view diffusion model, enabling the integration of information from multiple parts to create the final combined character. For training, we have contributed a large clothing-disentangled character dataset consisting of more than 10k anime characters. Extensive experiments demonstrate that our proposed approach not only facilitates efficient and high-quality disentangled 3D character generation with distinct clothing layers but also supports various cloth editing applications.

## 1 Introduction

The creation of 3D clothed character models is critically important across a range of applications, including film, augmented and virtual reality (AR/VR), and video gaming. In many applications, it is crucial to model the character's body and clothing separately, allowing users to freely change garments, thereby achieving controllable editing capabilities. However, manually creating such clothing-disentangled 3D characters is labor-intensive and time-consuming. Artists typically need to create individual components and then assemble them into complete clothed characters. Moreover, since the created clothing is often tailored to specific characters, it cannot be directly transferred to others, which may lead to issues such as penetration or misalignment. Therefore, automatically generating these clothing-disentangled character models from simple inputs (e.g., a single image) while enabling seamless clothing interchangeability presents a significant challenge.

Existing methods for creating clothing-disentangled 3D characters typically rely on optimization-based approaches. Based on input 3D avatar scans or textual descriptions of clothed avatars, these methods typically employ multi-layer 3D representations to model the body and each garment and then iteratively optimize these representations to fit the observations. The optimization leverages the powerful 2D diffusion model to implement Score Distillation Sampling (SDS) loss, which enables the disentangled reconstruction of clothing and the human body. Although these methods have achieved impressive results, they typically require multiple optimization processes to fully disentangle all clothing layers, with each optimization being quite time-consuming. This significantly limits their scalability and practicality.

In this paper, we propose the first feed-forward approach for efficiently generating clothing-disentangled 3D characters from a single image. The proposed approach achieves disentangled reconstruction significantly faster than existing optimization-based methods, reducing the process

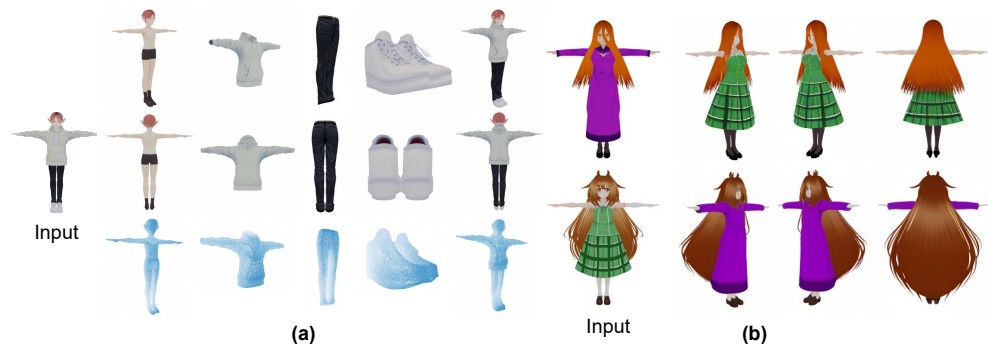

Figure 1: (a) Given a single image, this paper aims to generate clothing-disentangled 3D characters, supporting applications such as cloth transfer (b).

from several hours to mere seconds. We propose to first generate the multi-view images for each component of the clothed character, and then employ a generalizable multi-view reconstruction method to create the 3D models of each component.

However, generating high-quality, disentangled multi-view images remains challenging. Recent methods propose to introduce multi-view attention mechanisms into diffusion models, allowing them to generate multi-view images from a single input image. Although these methods achieve high-quality multi-view consistency, it remains unclear how to extend them to settings involving disentanglement. Naively incorporating additional part embeddings for disentanglement has been shown experimentally to result in poor part decomposition. Furthermore, many applications, such as virtual try-ons, require editing the clothing of characters clothing and then producing the edited character models, which necessitates the recombination of the decoupled components. Instead of editing the clothing in 3D space, which can lead to penetration and misalignment, we propose performing the editing at the image level. However, existing methods for component combination focus on settings with single-view images, rather than the multi-view image setting. Moreover, integrating a separate combination network tends to complicate and introduce redundancy into the entire pipeline.

To address this, we propose a novel two-stage disentanglement method that first disentangles each component in the 2D image space and then leverages the multi-view diffusion to obtain the multi-view images for each component. This design separates the tasks of component disentanglement and multi-view image generation, which simplifies each subtask and significantly enhances model performance. During 2D component disentanglement, our key insight is that different clothing parts are closely interconnected. Consequently, we propose a novel multi-part attention mechanism to enhance information exchange among different components, which significantly improves the quality of disentanglement. For component combination, rather than adding a separate combination network, we introduce a novel combination attention mechanism into the multi-view diffusion model that integrates information from multiple parts to generate the final combined character.

Existing datasets of clothing-disentangled characters typically contain only a limited number of subjects (fewer than 1,000) and offer restricted diversity and detail in clothing. To better train and evaluate the proposed model, we constructed a large clothing-disentangled character dataset, focusing on anime characters due to their abundant availability. This dataset comprises over 10,000 characters, with each character's body and clothing fully disentangled. Compared to existing datasets, these characters display more diverse and complex clothing styles, thereby posing greater challenges. Extensive experiments demonstrate that our proposed approach achieves efficient and high-quality clothing-disentangled 3D character generation, outperforming baseline methods.

In summary, this work makes the following contributions:

- We present the first novel framework for feed-forward 3D clothing disentangled avatar generation from a single image, which achieves high-quality results in a few seconds.
- We propose a two-stage disentanglement method that disentangles each component in the 2D image space and then generates the multi-view images for each part. We introduce a novel

multi-part attention mechanism for 2D component disentanglement and a combination attention mechanism for multi-view component combination.

- We construct the first large clothing-disentangled character dataset consisting of more than 10k anime characters, which will facilitate and inspire future research in this field.

## 2 RELATED WORKS

**Clothed human modeling.** In the initial stages of research on clothed human modeling, most works utilized parametric human meshes Bogo et al. (2016). These parametric models were enhanced with additional vertex offsets to achieve a more detailed and accurate representation of clothing Alldieck et al. (2018); Bhatnagar et al. (2019); Ma et al. (2020). Recent advancements in implicit functions Mescheder et al. (2019); Park et al. (2019); Mildenhall et al. (2020) have significantly advanced the development of techniques for reconstructing clothed humans from images Saito et al. (2019); Peng et al. (2021b;a); Dong et al. (2022); Saito et al. (2021); Tiwari et al. (2021); Wang et al. (2021b); Xiu et al. (2022); Chen et al. (2022), yielding impressive results. Furthermore, the recently proposed 3D Gaussian representations Kerbl et al. (2023) have significantly enhanced the rendering speed of NeRF-based clothed human reconstruction methods Zielonka et al. (2023); Qian et al. (2024); Xu et al. (2024), enabling rapid training and real-time rendering.

Although achieving remarkable results, these methods typically treat clothing and the human body as a unified entity, which hinders effective clothing manipulation and limits the range of applications. To achieve decoupled modeling of clothing and the body, some studies Yu et al. (2018a); Pons-Moll et al. (2017); Yu et al. (2019); Chen et al. (2021); Jiang et al. (2020); Hu et al. (2023); Corona et al. (2021); Kim et al. (2024); Moon et al. (2022); Dong et al. (2024); Feng et al. (2022) have introduced multi-layer human representations. Early works Jiang et al. (2020); Corona et al. (2021) extended parametric human body models by additionally learning parametric models of clothing from 3D clothing datasets. Leveraging these parametric models, some efforts Jiang et al. (2020); Moon et al. (2022) have achieved the reconstruction of coarse clothing shapes. However, due to the limited scope of 3D clothing datasets, these clothing models often lack diversity. Unlike parametric models of clothing, some methods Yu et al. (2018b) have proposed utilizing non-rigid surface tracking to fuse observations from RGB-D sequences, achieving high-quality reconstructions of clothing. Additionally, GALA Kim et al. (2024) has explored using the powerful priors of diffusion models to generate multi-layer 3D assets from a given single-layer clothed 3D human mesh. Others Zhang et al. (2023); Wang et al. (2023a); Dong et al. (2024) have even investigated generating multi-layer representations (e.g., NeRF and Mesh) directly from textual descriptions of clothing. Furthermore, Feng et al. (2022) proposed methods for reconstructing clothing disentangled human models from monocular videos using photometric loss. Although high-quality decoupled reconstructions have been achieved, these methods typically rely on per-scene optimization, requiring extensive optimization for each case, which limits their scalability and practicality. Unlike these optimization-based approaches, this paper aims to achieve rapid, feed-forward-based decoupled reconstruction.

**Diffusion model based 3D generation** Recent advancements in 2D diffusion models Ho et al. (2020); Rombach et al. (2022); Croitoru et al. (2023) and large-scale vision-language models such as CLIP Radford et al. (2021) have paved the way for novel methodologies in generating 3D assets, utilizing the robust priors established by these models. Innovative approaches DreamFusion Poole et al. (2023) utilize a per-scene optimization scheme where a 2D text-to-image generation model is distilled to produce 3D models directly from textual descriptions. This paradigm has been further explored in numerous recent studies Chen et al. (2023a); Wang et al. (2023d); Seo et al. (2023a); Yu et al. (2023); Lin et al. (2023); Zhu & Zhuang (2023); Huang et al. (2023a); Seo et al. (2023b); Tsalicoglou et al. (2023); Armandpour et al. (2023); Chen et al. (2023c). Complementary research has expanded these techniques to facilitate image-to-3D synthesis Tang et al. (2023); Melas-Kyriazi et al. (2023); Qian et al. (2023); Xu et al. (2022); Raj et al. (2023); Shen et al. (2023). Unlike generating a single holistic object, recent works Li et al. (2023); Cheng et al. (2023) propose methods for generating objects comprised of multiple parts, by additionally optimizing the relative positions of these parts. Although significant progress has been made, these methods typically utilize a Score Distillation Sampling (SDS) loss to optimize 3D representations (such as NeRF, 3D Gaussian, or mesh), which results in inefficiencies that limit scalability.

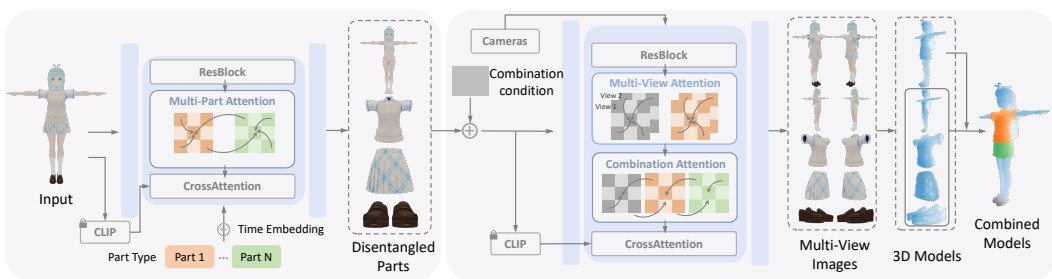

Figure 2: **Overview of the proposed method.**

In contrast, some studies Nichol et al. (2022); Zeng et al. (2022); Wang et al. (2023c); Jun & Nichol (2023) propose extending diffusion models to 3D, directly outputting representations such as point clouds and neural fields, thereby enabling efficient 3D generation. However, due to the limited availability of 3D data, the generalization capability of these methods remains relatively constrained.

Alternatively, another line of research focuses on extending a pretrained 2D diffusion model to initially generate multi-view images Liu et al. (2023b;c); Huang et al. (2023b); Shi et al. (2023); Long et al. (2023), followed by multi-view 3D reconstruction based on these images Liu et al. (2023a); Long et al. (2023); Hong et al. (2023); Wang et al. (2023b); Tang et al. (2024). Pioneering work zero123 Liu et al. (2023b) introduces relative view conditions to diffusion models, enabling novel view synthesis from a single image. To improve multi-view consistency, recent works propose to introduce multi-view attention to the diffusion models. SyncDreamer Liu et al. (2023c) introduces a 3D global feature volume to fuse multi-view information, while EpiDiff Huang et al. (2023b) incorporates epipolar constraints to facilitate efficient cross-view interactions among feature maps from neighboring views. In addition, MVDream Shi et al. (2023) and Wonder3D Long et al. (2023) propose to reuse the self-attention layers for multi-view interaction. Although these methods have achieved impressive results, they typically generate multi-view images of a single holistic object and are unable to decouple and generate images of multiple distinct parts. In contrast, this paper proposes a novel disentanglement method that supports the generation of multi-view images for each clothing part of a clothed human. Based on the generated multi-view images, early works propose to reconstruct 3D models with the per-scene optimization scheme such as neural surface reconstruction Wang et al. (2021a) and SDS loss based optimization. To improve the efficiency, recent works Liu et al. (2023a); Tang et al. (2024); Peng et al. (2024) propose to directly reconstruct 3D models from multi-view images using feed-forward neural networks. Training on large scale datasets Deitke et al. (2023), these methods can achieve high-quality 3D reconstruction in seconds.

## 3 METHOD

We aim to generate the clothing disentangled 3D character models from a single image. In contrast to optimization-based methods, this paper proposes a novel feed-forward approach that not only enables efficient clothing disentangled character generation in seconds but also intrinsically supports 3D virtual try-on applications without the need for additional clothing combination networks. Figure 2 presents an overview of our approach. Given an input image of the clothed character, a multi-part diffusion model is proposed to produce disentangled images of the human body and each clothing part (Section 3.1). The disentangled images are then fed into a multi-view diffusion model to generate multi-view images for each part. To compose the body and clothing parts, a novel multi-part attention module is introduced to guide the composition process (Section 3.2). Based on these multi-view images, the off-the-shelf feed-forward multi-view reconstruction methods are adopted to produce the 3D model of each part, and an optional optimization method is proposed to combine all 3D models (Section 3.3).

## 3.1 2D CLOTHING DISENTANGLEMENT

To disentangle the human body and each clothing part from the input image, existing methods typically rely on clothing segmentation (i.e., semantic segmentation), which outputs the pixel-wise labels of each type. However, the clothing segmentation based methods have two main limitations. First, the segmented image of each type is often incomplete due to the occlusion. Second, the obtained part images usually only occupy a small part of the original image, resulting in a low resolution. To address these issues, we propose a novel 2D part disentanglement method based on diffusion models, which can generate complete and high-resolution part images from the input image. Before introducing our approach in detail, we first briefly describe the diffusion model.

**Diffusion models.** Diffusion models (Sohl-Dickstein et al., 2015; Ho et al., 2020) are a category of generative models parameterized by Markov chains, comprising forward and reverse processes. In the forward process, a sample $x_0$ drawn from the data distribution $p(x)$ is progressively noised through a sequence $\{x_t \mid t \in (0, T)\}$, where each $x_t$ is computed as $x_t = \alpha_t x_0 + \sigma_t \epsilon$ and $T$ denotes the number of train steps. Here, $\epsilon$ represents random noise sampled from a normal distribution $\mathcal{N}(0, 1)$, and $\alpha_t, \sigma_t$ are constants defining the noise schedule, culminating in complete Gaussian noise. Conversely, the reverse process iteratively denoises the noisy image, reconstructing $x_{t-1}$ from $x_t$ by estimating the noise $\epsilon$. This estimation is achieved by a noise predictor $\epsilon_\theta$, parameterized using a UNet architecture. Please refer to Ho et al. (2020); Sohl-Dickstein et al. (2015) for more details about diffusion models.

Under the diffusion model scheme, our goal to disentangle each part from the input image $\mathbf{y}$ can be formulated as follows:

$$f(\mathbf{y}) = p\left(\mathbf{x}_T^1, ..., \mathbf{x}_T^N\right) \prod_{t=1}^{T} p_{\boldsymbol{\theta}}\left(\mathbf{x}_{t-1}^1, ..., \mathbf{x}_{t-1}^N \mid \mathbf{x}_t^1, ..., \mathbf{x}_t^N, \mathbf{y}\right) \tag{1}$$

where $\mathbf{x}_T^1, ..., \mathbf{x}_T^N \sim \mathcal{N}(\mathbf{0}, \mathbf{I})$ and $N$ denotes the number of parts. Therefore, the key problem is to calculate the distribution $p_{\boldsymbol{\theta}}$, which enables to generate the disentangled part images based on the Markov chain.

To leverage the strong image prior learned from billions of images, we base our model on the pretrained stable diffusion models Ho et al. (2020). The original diffusion model only supports single image generation based on the conditional image. To enable multi-part image generation, we additionally introduce the part type $\mathbf{c}$ condition into the stable diffusion models, and then the diffusion model can generate a specified part image based on the input image and the corresponding part type, i.e., $\mathbf{x}^{\mathbf{c}} = f(\mathbf{y}, \mathbf{c})$. Specifically, we adopt the one-hot encoding to represent the part type, which is further augmented with the positional encoding and then concatenated with the time embedding. Intuitively, using the multi-part diffusion model to generate the different part images separately will lose the strong correlation among different parts since there are fixed patterns in how people wear their clothes. To address this, we introduce a novel multi-part attention module to facilitate information propagation across different parts, implicitly encoding multi-part dependencies.

Rather than adding a new layer, we achieve this by extending the original self-attention layers to be multiple parts aware, which allows connections to other parts within the attention layers. This module not only enhances the part decomposition quality but also achieves faster convergence. The specific calculation of queries, keys, and values of part $\mathbf{c}$ in the multi-part attention layer is as follows:

$$\mathbf{q}^{\mathbf{c}} = \mathbf{Q} \cdot \mathbf{z}^{\mathbf{c}}, \mathbf{k}^{\mathbf{c}} = \mathbf{K} \cdot (\mathbf{z}^{\mathbf{1}} \oplus \cdots \oplus \mathbf{z}^{\mathbf{N}}), \mathbf{v}^{\mathbf{c}} = \mathbf{V} \cdot (\mathbf{z}^{\mathbf{1}} \oplus \cdots \oplus \mathbf{z}^{\mathbf{N}}) \tag{2}$$

where $\mathbf{Q}$, $\mathbf{K}$ and $\mathbf{V}$ denote query, key and value embeddings matrices, $\mathbf{z}^{\mathbf{c}}$ denotes the latent embeddings of part $\mathbf{c}$ in transformer blocks, and $\oplus$ denotes concatenation operation.

## 3.2 MULTI-VIEW GENERATION AND PART COMBINATION

To reconstruct the 3D models of each part, we propose to first generate the multi-view images based on the disentangled image, which can be used as the input of the multi-view based reconstruction methods. To achieve this, similar to the previous works, we introduce the multi-view attention layers to the diffusion models to enhance the multi-view consistency. Specifically, we modify the original

self-attention layers to be multi-view aware, which allows information propagation across different views within the attention layers. With this design, the multi-view diffusion model can generate multi-view consistent images for each part.

In addition to generating multi-view images for each part, the composition of parts is also an indispensable component for many applications, such as virtual try-on. Existing 2D part composition methods typically train a dedicated network to combine the images of different parts. These methods usually are designed for single view images and how to extend them to multi-view images is unknown. Moreover, adding an additional network to handle part composition increases the overall complexity and redundancy of the method.

Thus, in this paper, we propose to introduce a part composition module into the multi-view diffusion model to achieve efficient and high-quality multi-view part composition. Specifically, we introduce a combination attention module after the multi-view attention layer in the UNet to guide the combination process, which can exchange the information of multiple parts. To generate the combined images, an alternative method is directly fusing the information of multiple parts from the corresponding conditioned part images. However, as shown in the experiments, this design can not achieve high-quality combination since the network needs to simultaneously learn multi-view generation and part combination from the input part images. To address this, we propose to introduce a special condition image specifically for part combination. Intuitively, the network learns multi-view generation from input part images and part combination from special condition images, thereby achieving improved performance. The calculation of queries, keys, and values of each part is similar to Equation 2.

### 3.3 MULTI-VIEW BASED RECONSTRUCTION

Given the multi-view images of each part, we can adapt the recent feed-forward multi-view based reconstruction methods to produce the 3D models of each part. In particular, we adopt the LGM model to reconstruct the 3D models of each part, which can produce high-quality 3D gaussian models in 1 second.

In addition, since the reconstructed 3D model of each part is part centric, the mutual position relationship between different parts is not considered. To address this, we propose an optional 3D part model optimization algorithm to optimize the mutual position relationship between different parts. The key idea is to leverage multi-view rendering consistency between the rendered images of combined 3D Gaussian models and part combination images from multi-part attention modules to optimize the mutual position relationship between different parts. Specifically, for each part $\mathbf{c}$, we aim to produce the corresponding rotation matrix $\mathbf{R_c}$, translation vector $\mathbf{T_c}$, and scale factor $s_\mathbf{c}$ of 3D gaussian model $\mathbf{p_c}$ relative to the unified canonical space, which can be written as follows:

$$\min_{\mathbf{R},\mathbf{T},\mathbf{s}} \sum_{\mathbf{v}} \|D_v(f_{cat}(s_i(\mathbf{R}_i\mathbf{p}_i + \mathbf{T}_i)) - \mathbf{I}_v\|_2, \tag{3}$$

where $\mathbf{R}, \mathbf{T}, s$ denotes the collections of corresponding $\mathbf{R_c}, \mathbf{T_c}, s_\mathbf{c}$, $D_v(\cdot)$ denotes the differentiable 3D gaussian rendering, and $\mathbf{I}_v$ denotes the part combination images from multi-part attention module of view $v$.

## 4 EXPERIMENTS

### 4.1 DATASET AND METRICS

Existing 3D character datasets often suffer from limitations such as the integration of clothing with character models, making them inseparable, or they are too small (fewer than 1,000 models) with overly simplistic and non-diverse clothing options. Inspired by the PAniC3D Chen et al. (2023b), we collected a comprehensive dataset of over 13,000 anime characters from VRoidHub VRoid (2022), which contains rich diversity and complexity in character designs. To facilitate flexible usage, we developed a robust rendering pipeline that controls the visibility of specific parts of the character models, namely body, upper body clothing, lower body clothing, and shoes. This approach allows for precise manipulation of different clothing combinations within the dataset. Upon manual review and removal of incorrectly disentangled clothing models, our refined dataset comprises over 10,000

Table 1: Quantitative comparisons of disentangled multi-view image generation.

|          | PSNR ↑   | SSIM ↑  | LPIPS ↓ |
|----------|----------|---------|---------|
| Baseline | 17.5499  | 0.7637  | 0.2394  |
| Ours     | **26.6920** | **0.9119** | **0.0737** |

character models. Please refer to the appendix for details. Each of these models is capable of generating images in 11 distinct clothing combinations. We reserve 500 character models for evaluation with the remaining models used for training. Ultimately, our dataset expanded to encompass more than 110,000 character models with unique clothing. For each model, we rendered images from four distinct perspectives to generate multi-view images for training and evaluation.

For the evaluation of image synthesis, we employ three metrics: Peak Signal-to-Noise Ratio (PSNR), Structural Similarity Index (SSIM), and Learned Perceptual Image Patch Similarity (LPIPS) Zhang et al. (2018).

### 4.2 IMPLEMENTATION DETAILS

The Stable Diffusion Image Variations Model is utilized as the basis for both our 2D part disentanglement and multi-view generation and combination models. The optimizer settings and $\epsilon$-prediction strategy are consistent with those used during the training of the Image Variations Model. In stage one, we employ a batch size of 512 and train 20,000 steps. In stage two, we use a batch size of 600 and train 30,000 steps. The whole training process typically spans approximately three days, utilizing a cluster of eight Nvidia Tesla A100 GPUs.

### 4.3 COMPARISONS WITH THE BASELINE METHOD

Existing methods for reconstructing objects from single images generally support only holistic reconstruction. To facilitate a comparison with our approach, we have extended these existing holistic reconstruction methods to enable multi-part decoupling and generation. Specifically, we enhanced the state-of-the-art method, Wonder3D Long et al. (2023), by removing irrelevant normal modules and incorporating 'part type' as an additional condition. This adaptation allows for the generation of multiple parts, making it possible to compare directly with our method.

We evaluate the quality of generated multi-view images for each part. Tab. 1 presents the quantitative comparisons between our method and the baseline method. As shown in the table, our approach outperforms the baseline across all metrics. We also present qualitative results in Figure 3, which shows that the propose two-stage pipeline achieves greater consistency with the input image, and avoids the type mispredictions often seen with the baseline approach.

### 4.4 ABLATION STUDIES

We conduct ablation studies to validate the effectiveness of the design in our proposed method.

**Multi-part attention for 2D part disentanglement.** We explore the effectiveness of the multi-part attention module in our 2D part disentanglement framework. An alternative approach disables the multi-part attention module, thereby generating each part independently without leveraging cross-part contextual information. The quantitative results for part disentanglement are reported in Tab. 2, which shows that our approach with the multi-part attention module significantly outperforms the baseline. The qualitative results are illustrated in Figure 4, which demonstrates that the multi-part attention module significantly enhances the quality of disentangled part images.

**Multi-view part composition.** We propose to introduce a special condition image and use the combination attention mechanism to guide the part combination as shown in Section 3.2. An alternative approach is to first train a multi-view diffusion model and then use the attention mechanism to fuse the multi-part feature based on only part images rather than introducing another condition image. The quantitative and qualitative results for part composition are shown in Tab. 1 and Figure 5 (left), respectively. As we can see, our approach with the special condition image significantly outperforms the baseline method.

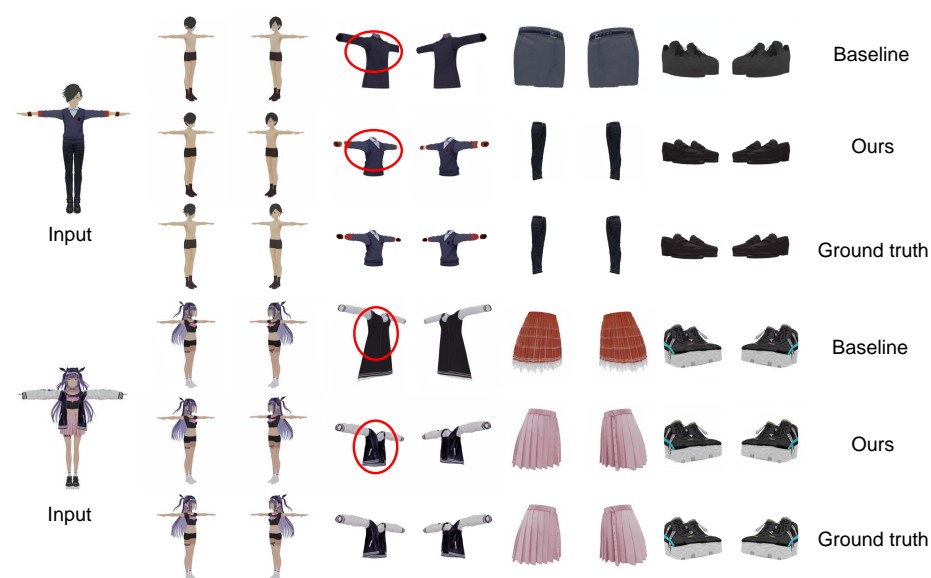

Figure 3: **Qualitative comparisons of disentangled multi-view image generation.**

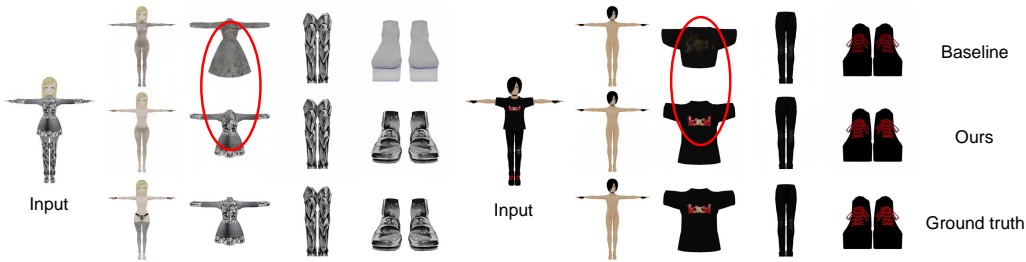

Figure 4: **Qualitative comparisons for 2D part disentanglement.**

**3D part model composition.** We propose an optimization-based method to fuse 3D part models to obtain a complete 3D character model. We compared it with the 3D models generated directly from the combined images. The qualitative results are shown in Figure 5 (right), where we compare the quality of rendered images from the 3D models. As the results show, our optimization-based method achieves better rendering quality than the baseline approach. The disentangled part images can capture much more details than the composed image, which produces more detailed 3D models of each part. This highlights the value of disentangled generation.

Table 2: Ablation studies for 2D part disentanglement and part composition.

|                            | PSNR ↑  | SSIM ↑ | LPIPS ↓ |
|----------------------------|---------|--------|---------|
| w/o multi-part attention   | 25.7860 | 0.8899 | 0.0974  |
| w/ multi-part attention    | **27.3969** | **0.9098** | **0.0774**  |
| w/o special condition image| 16.8848 | 0.8317 | 0.1516  |
| w/ special condition image | **25.2128** | **0.9320** | **0.0583**  |

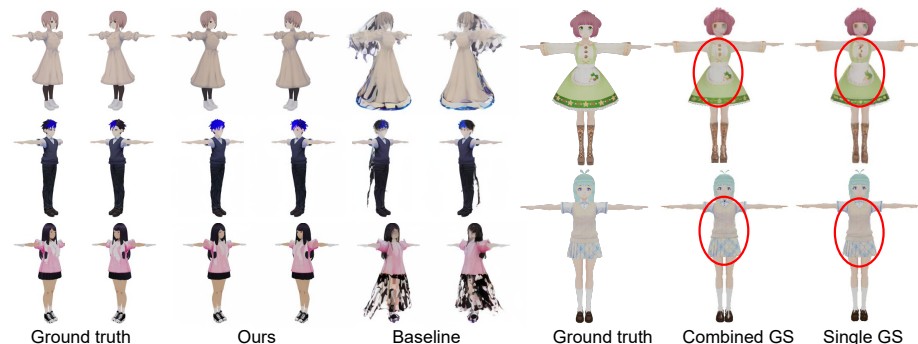

Figure 5: **Results for multi-view part composition and 3D part model composition. 'Single GS' represents the single Gaussian representation of the entire character.**

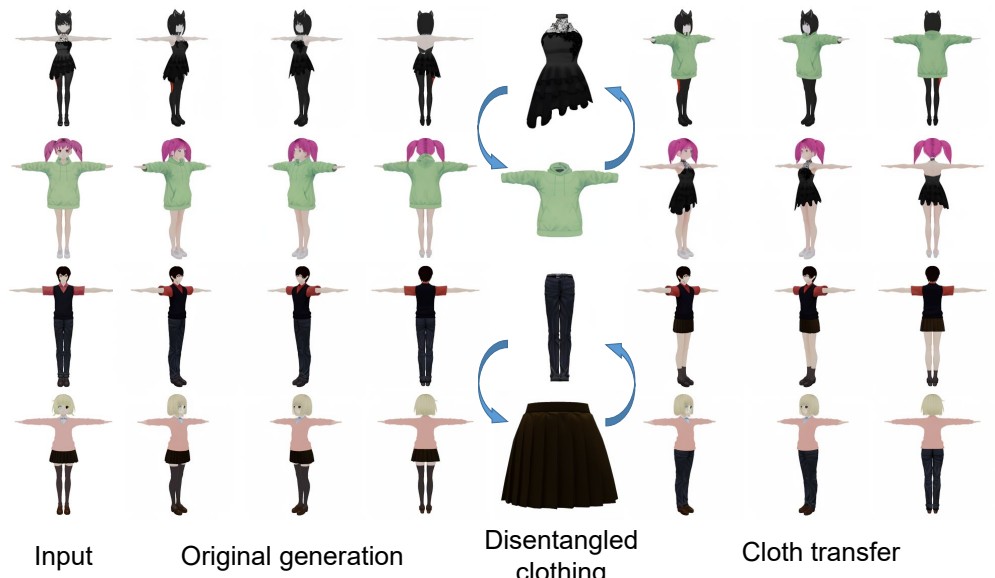

Figure 6: **Application for the cloth transfer.**

## 4.5 APPLICATIONS

Thanks to the design of disentangled modeling and part composition, our method naturally supports 3D virtual try-on applications. We present the clothing transfer results in Figure 6. Furthermore, the 3D models generated by our method can be adapted to fit a parametric human model, which enables dynamic animation of reconstructed models. The results are shown in Figure 7 (bottom).

## 4.6 LIMITATIONS

The proposed approach still has the following limitations. Although we have contributed a large clothing-disentangled 3D dataset, its size remains insufficient compared to existing image datasets. Therefore, exploring how to train disentangled models using 2D image data is our next step. Additionally, our method generates from a single image, resulting in a static 3D clothing model, which limits the ability to create realistic animations of the clothing models. Thus, decoupling dynamic clothing models from sequential inputs is left as future work.

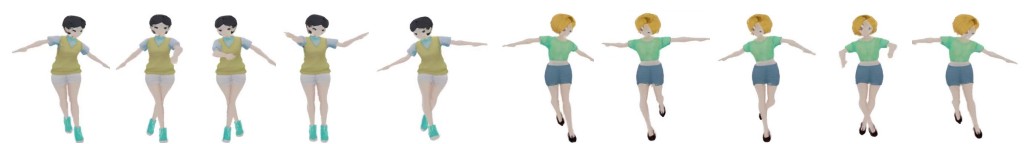

Figure 7: **Application for animation.**

## 5 CONCLUSION

In this paper, we propose a novel method for generating clothing-disentangled 3D characters from a single image. Unlike previous optimization-based methods, we introduce a feed-forward decoupling reconstruction approach that initially generates multi-view images of each part, followed by individual reconstruction using a generalized multi-view reconstruction method. To achieve high-quality decoupling, we propose a two-stage method that first decouples each part in the 2D image space, and then generates multi-view images for each part. A novel cross-part diffusion model is introduced to enhance information exchange among different components during the 2D component decoupling stage. To generate the final combined character, we introduce a novel combination attention mechanism into the multi-view diffusion model that integrates information from multiple parts. For training, we have created a large dataset of more than 10k clothing-disentangled characters. Experiments show that our method achieves efficient, high-quality generation of clothing-decoupled 3D characters, while also supporting clothing editing applications.

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

## 6 APPENDIX

In this appendix, we provide more details.

In the first part, we show some examples of our dataset. Our dataset expanded to encompass more than 110,000 character models with unique clothing. The dataset includes a diverse range of clothing and shoe types, which can be used for tasks such as garment generation and editing.

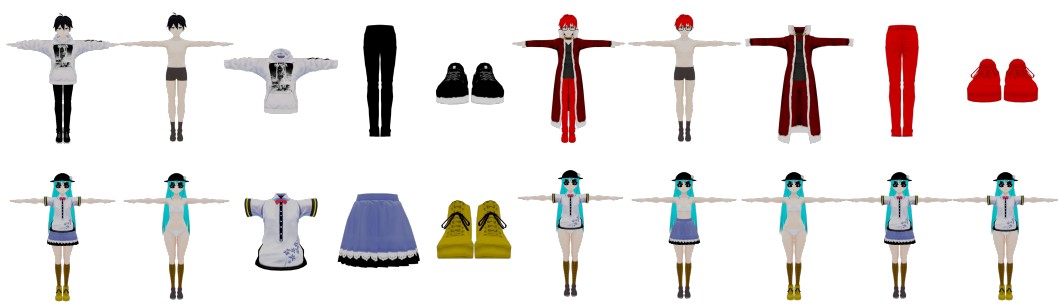

Figure 8: **Examples of the dataset.**

Note that we also include a short video showcasing the qualitative results of our pipeline in the Supplemental Material.

In the second part, we provide details about the animation. While the primary focus of this paper is on clothing-disentangled generation, we have implemented a straightforward animation pipeline to animate the generated 3D characters. Specifically, we begin by using OpenPose Cao et al. (2019) to estimate the 2D human poses from the generated multi-view images of the 3D character. Subsequently, we fit a parametric human model (SMPL Bogo et al. (2016)) to these estimated multi-view 2D poses. To animate the reconstructed 3D Gaussian characters, we identify the closest vertex on the SMPL model for each Gaussian point and utilize the corresponding skinning weight of that vertex for animation. Finally, we animate the 3D Gaussian characters using the SMPL motion parameters. Exploring more sophisticated animation methods will be a focus of our future research.

In the third part, we provide the details of the experiments. Following previous works Long et al. (2023); Shi et al. (2023), we use 256x256 resolution for both generated disentangled 2D cloth images and multi-view images. The combination condition is a predefined constant matrix, which matches the shape of the disentangled images and has all its values set to 128. This matrix serves as a guide for the combination of different parts in the diffusion model. The total number of generated images is calculated as $(N_{part} + 1) * N_{view}$, where $N_{part}$ represents the number of parts, and $N_{view}$ represents the number of views. The additional one denotes the generated multi-view images of the combined character. The additional term accounts for the generated multi-view images of the combined character. In our specific setup, with $N_{part} = 4$ and $N_{view} = 4$, we produce a total of 20 images. The number of body parts, being four, is fixed throughout our experiments. The optimization process of Section 3.3 typically takes around 2 minutes.

