# OpenReview forum: "Clothing-disentangled 3D character generation from a single image"
_ICLR.cc/2025/Conference — Submitted to ICLR 2025_

### Official Review · Reviewer_WHuA · 2024-10-15

**Soundness:** 3
**Presentation:** 2
**Contribution:** 2
**Rating:** 6
**Confidence:** 3

**Summary:**

The paper introduces a method for generating clothing-disentangled 3D anime characters from a single image. In the first stage, the model takes a frontal image of a clothed anime character in a canonical pose and generates images of the top, bottom, shoes, and minimally dressed body using a stable diffusion model. In the second stage, the diffusion model is applied to generate four distinct views for each part, conditioned on the outputs from the first stage. Finally, the 3D structure is produced using the existing LGM method by taking the generated multi-view images as input, which generates a 3D Gaussian asset. The work has also proposed a synthetic dataset for training the model, consisting of 110k anime characters generated from VRoid.

**Strengths:**

1.The paper proposes using a diffusion model as a partitioning method to avoid occlusion and low resolution issues.

2.The model is feed-forward and does not require an optimization process, enabling the generation of 3D assets in a short amount of time.

3.The model allows for anime character cloth-switching and supports virtual try-on applications.

4.The dataset includes a rich variety of anime characters with diverse outfits.

**Weaknesses:**

1.The technical contribution appears limited, as the main novelty lies in fine-tuning the diffusion model. The reconstruction is performed using an existing method, so the paper primarily focuses on generating multi-view images for each part of the characters.

2.The generated assets are represented as 3D Gaussian splats, which may not be as practical as mesh-based models for downstream applications due to the lack of geometric detail.

3.In the dataset samples shown, while there is variation in the outfits, the body shapes appear to lack diversity. This raises a concern about the model's ability to handle virtual try-on between characters with significantly different body shapes (e.g., slim vs. thick body types).

**Questions:**

1. L253-254: The model seems to generate one part of the entire image at a time, correct? This generation is conditioned on the specific part type provided. Could you clarify if this understanding is accurate?

2. L287: Could you explain what is meant by "this special condition image"? A more detailed explanation of how it works would be helpful for understanding this part of the method.

3. L295: The citation for the LGM method appears to be missing.

---

> ### Author Response · Authors · 2024-11-25
>
> ### Technical Contribution
>
> 1. We respectfully disagree with the opinion that the "main novelty lies in fine-tuning the diffusion model." As demonstrated in our experiments (Table 1 and Figure 3 in the original manuscript), directly fine-tuning the diffusion model (i.e., the baseline method) struggles to correctly disentangle individual part images or combine them into a coherent clothed human image. These capabilities are critical for applications such as virtual try-on.
> 2. This paper introduces a novel task: generating disentangled multi-view images for each part of a character from a single input image. This task is fundamentally different from holistic clothed human generation. To address this challenge, we propose a novel two-stage disentanglement method that first disentangles each component in the 2D image space and then generates multi-view images for each part.
> 3. Specifically, we introduce a **multi-part attention mechanism** for 2D component disentanglement and a **combination attention mechanism** for multi-view component integration, enabling high-quality part disentanglement and seamless clothing combination.
> 4. Furthermore, we construct the first large-scale clothing-disentangled character dataset to support future research in this area.
> 5. Taken together, we believe the contributions of this paper are substantial and address a significant gap in the field.
>
> ### 3D Gaussian Splats
>
> 1. As discussed earlier, the primary focus of this paper is on generating disentangled multi-view images. For the multi-view reconstruction, we adopt an existing method, LGM, which supports high-quality rendering generation. While LGM outputs 3D Gaussian splats, other methods such as [a] and [b], which produce mesh-based representations, can also be employed within our framework if desired.
>
> ### Body Shape Diversity
>
> 1. Intuitively, our method does not rely on predefined body shape priors but instead learns the combination in a data-driven manner. As a result, our approach is capable of handling significant differences in body shapes. In the revised manuscript, we will include additional virtual try-on results showcasing the model's ability to manage large variations in body shapes.
>
> ### Question 1
>
> 1. Yes, our model generates one part of the entire image at a time, conditioned on the specific part type provided.
>
> ### Question 2
>
> 1. Without the combination attention module, the multi-view generation model is limited to generating multi-view images for each individual part separately. To address this, we integrate the combination attention module into the model, enabling it to combine these individual parts into a unified clothed human. For each batch, the original condition input includes only the images of individual parts. However, our goal is to generate multi-view images not only for each part but also for the combined clothed human. To achieve this, we introduce a special combination condition image as a flag, which signals the model to generate the multi-view images of the combined clothed human. As demonstrated in our experiments, this design significantly improves the quality of the combined results, producing more cohesive and realistic clothed human models.
>
> ### Question 3
>
> 1. We will include the citation in the revised manuscript.

---

### Official Review · Reviewer_W2Hv · 2024-10-29

**Soundness:** 3
**Presentation:** 2
**Contribution:** 2
**Rating:** 6
**Confidence:** 4

**Summary:**

The goal of this study was to perform 3D reconstruction by separating clothing and body components from single character image, utilizing a diffusion model to generate 2D cloth disentanglement and multi-view images. The LGM model was used for 3D reconstruction, and a dataset of 10k 3D character models was built.

**Strengths:**

This study extends existing image-to-3D techniques using diffusion models to achieve cloth disentanglement.
Additionally, by building a 10k 3D character dataset, which, if released, could greatly benefit for future research.

**Weaknesses:**

1. The methods used for clothing disentanglement and 3D reconstruction are not novel, primarily involving applications of existing methods. The diffusion model is the similar to Stable Diffusion [1], the attention method is the similar to Animate-Anyone [2], and the 3D reconstruction method is the similar to LGM[3].

2. Throughout the paper, there is a lack of references when discussing specific methods or abbreviations (e.g line 43,47,74,78,294,295 and more), which can make it difficult to follow.

3. The comparative analysis is limited to a baseline, lacking detailed explanations of differences, making it hard to understand why the qualitative and quantitative results differ.

4. It’s disappointing that there are no experiments using general character images with various poses and perspectives, which limits the practical applicability.

[1] Robin Rombach, Andreas Blattmann, Dominik Lorenz, Patrick Esser, Björn Ommer. High-Resolution Image Synthesis with Latent Diffusion Models.

[2] Li Hu, Xin Gao, Peng Zhang, Ke Sun, Bang Zhang, Liefeng Bo. Animate Anyone: Consistent and Controllable Image-to-Video Synthesis for Character Animation.

[3] Jiaxiang Tang, Zhaoxi Chen, Xiaokang Chen, Tengfei Wang, Gang Zeng, and Ziwei Liu. Lgm:
Large multi-view gaussian model for high-resolution 3d content creation.

**Questions:**

1. Did you use a pretrained stable diffusion model for fine-tuning?

2. Why didn’t you use 3D evaluation metrics such as Chamfer Distance or Point-to-Surface?

3. Is there optimal number of views or view point for 3D reconstruction?

---

> ### Author Response · Authors · 2024-11-23
>
> ### Applications of Existing Methods
>
> 1. **Use of Stable Diffusion Architecture**
>
>     We do not believe that adopting a similar architecture to Stable Diffusion [1] diminishes the novelty of our work. Many influential papers [a][b][c] have also adopted similar architectures, demonstrating that the framework is a robust foundation for diverse applications. Our contributions lie in addressing the novel task of clothing disentangled generation.
>
> 2. **Attention Module**
>
>     We respectfully disagree with the assertion that our proposed attention module is similar to that of Animate-Anyone [2]. In Animate-Anyone, the attention module is designed to model temporal interactions. In contrast, our attention mechanism serves two distinct purposes: (1) enhancing information exchange between different body parts in Stage 1 and (2) combining these parts into a unified clothed human in Stage 2. These approaches address entirely different problems. Moreover, our method is the first to generate disentangled multi-view images for each part from a single image, which we believe establishes its novelty.
>
> 3. **Multi-View Reconstruction Model**
>
>     Regarding the multi-view reconstruction model, we directly utilize an existing method, LGM [3], as an off-the-shelf solution. We do not claim this aspect as part of our contributions, as the focus of our work lies elsewhere.
>
> ### Lack of References
>
> 1. Thank you for the reviewer’s valuable feedback. While we provide detailed explanations in Section 2, we acknowledge the need for more specific references in the mentioned lines. We will include the corresponding references in the revised manuscript to improve clarity and readability.
>
> ### Comparative Analysis
>
> 1. As discussed in the paper, the task of **clothing-disentangled generation** from a single image is a novel problem, and most existing methods are designed solely for generating a holistic clothed human. To the best of our knowledge, multi-view diffusion models hold the potential to be extended to this new task. Therefore, we adapted the state-of-the-art multi-view diffusion method as our baseline, enabling it to effectively handle multi-part decoupling and generation.
> 2. We have provided a detailed explanation of the baseline method in Lines 350–356. Our proposed method and this baseline are fundamentally different approaches, targeting the same task from distinct perspectives.
>
> ### Various Poses and Perspectives
>
> 1. This paper represents the first attempt to address the novel task of **clothing-disentangled generation** from a single image. It is impractical for a single paper to tackle all the challenges in this new domain. The main contributions of this work are:
>
>     a) proposing a new task focused on disentangled generation,
>
>     b) developing the first method capable of achieving high-quality disentangled results, and
>
>     c) constructing a large-scale dataset to support this research.
>
>     While handling various poses and perspectives is not the primary focus of this paper, we believe our work lays a solid foundation for future research to build upon.
>
> 2. Additionally, existing studies [d][e] have shown that recovering T-pose or A-pose representations from images with diverse poses and perspectives is relatively straightforward. These findings indicate that such challenges can be addressed in future work or by integrating existing techniques.
>
> ### Question 1
>
> 1. Yes, similar to previous works [a][b][c], our model leverages the pretrained weights of the Stable Diffusion model.
>
> ### Question 2
>
> 1. The output 3D model in our method is represented by **3D Gaussians**, which are fundamentally different from traditional mesh representation. Metrics such as Chamfer Distance or Point-to-Surface are not suitable for evaluating this type of representation.
>
> ### Question 3
>
> 1. The multi-view reconstruction module is not the primary focus of this paper. We directly adopt the LGM model, which utilizes four views for reconstruction. Further exploration of the optimal number of views could be an interesting direction for future work.
>
> [a] Zero-1-to-3: Zero-shot one image to 3d object
>
> [b] Mvdream: Multi-view diffusion for 3d generation
>
> [c] Wonder3d: Single image to 3d using cross-domain diffusion
>
> [d] Charactergen: Efficient 3d character generation from single images with multi-view pose canonicalization
>
> [e] Adding Conditional Control to Text-to-Image Diffusion Models

---

> > ### Comment · Reviewer_W2Hv · 2024-11-26
> >
> > Thank you for your response.
> > I had some doubts about how well the suggested method would work in general images, but I agree that combining it with the additional methods you mentioned is a viable consideration. Also, when I mentioned that the concatenation method in the attention mechanism is similar to how Animate Anyone conditions the source image, I was referring to that comparison. However, I find it particularly interesting and meaningful that this paper used it as a multi-view condition. Therefore, I will increase the score.

---

### Official Review · Reviewer_9wsZ · 2024-11-01

**Soundness:** 3
**Presentation:** 2
**Contribution:** 2
**Rating:** 6
**Confidence:** 4

**Summary:**

Given an input image of an anime character, this paper presents a method to reconstruct the character in 3D where the body and the clothing items are disentangled. This is achieved by first generating complete images of each clothing item (assuming a fixed set of clothing types) from the input image. Then multiview images of each clothing item are generated which are used for 3D reconstruction. Finally, the different 3D generations are fit together with an optimization.

**Strengths:**

- In the context of human/character reconstruction, going from a single image to a disentangled body/clothing 3D representation is important and this paper makes good observations without necessarily relying on parametric body models.

- Some aspects of the method design choices are ablated well.

**Weaknesses:**

- Even after reading several times, I do not actually fully understand what the combination attention and combination condition image is doing. It's especially confusing since the appendix says: "The combination condition is a predefined constant matrix, which matches
the shape of the disentangled images and has all its values set to 128." If this matrix is fixed with all values equal to 128, what does it actually do?
- There is not a lot of evaluation on how well the method generalizes. It is not easy to tell how similar are the testing images to the training ones. Also maybe testing on some out of domain images would be useful.

**Questions:**

- I'm not sure how the baseline of Wonder3D is run actually. The authors mention they add a "part" condition. Do they still use their part decomposed images as input image or do they use the full character image as the input and generate multiview images of each part?
- Once each part is reconstructed as 3D gaussians, it seems the paper tries to optimize the transformation of each part and then just directly overlay the Gaussians. Do they do anything related to the opacity values of the Gaussians? If not, wouldn't the rendering of the composite would look different then the rendering of the individual parts?

---

> ### Author Response · Authors · 2024-11-21
>
> ### Combination Attention and Combination Condition Image
>
> 1. Without the combination attention module, the multi-view generation model is limited to generating multi-view images for each individual part separately. To address this, we integrate the combination attention module into the model, enabling it to combine these individual parts into a unified clothed human. For each batch, the original condition input includes only the images of individual parts. However, our goal is to generate multi-view images not only for each part but also for the combined clothed human. To achieve this, we introduce a special **combination condition image** as a flag, which signals the model to generate the multi-view images of the combined clothed human. As demonstrated in our experiments, this design significantly improves the quality of the combined results, producing more cohesive and realistic clothed human models.
>
> ### More Evaluation
>
> 1. Based on our observations, the test images include characters and clothing types that were never seen in the training data. Additionally, the test set features many clothing styles that differ significantly from those in the training data. We appreciate the reviewer’s valuable suggestion. In the revised manuscript, we will include additional results on out-of-domain images.
>
> ### Question 1
>
> 1. The baseline of Wonder3D uses the full character image as the input. It is important to note that the decomposed part images are generated as outputs of our method in Stage I.
>
> ### Question 2
>
> 1. We do not adjust the opacity values of the Gaussians during the optimization process. Once the spatial positions of each part are accurately recovered, the rendering of the composite Gaussians will appear consistent with the rendering of the individual parts.

---

> > ### Comment · Reviewer_9wsZ · 2024-11-26
> >
> > Thank you for the detailed answers. I think I have a better understanding of how combination condition image works, it seems it is more like a switch. While I still have some reservations about the generalization of the method, given that the authors are willing to include out of domain examples in their final version, I'm okay to accept this paper. It provides useful ideas in terms of how to provide separate reconstructions of the body and the clothing.

---

### Official Review · Reviewer_HfJ1 · 2024-11-04

**Soundness:** 2
**Presentation:** 2
**Contribution:** 2
**Rating:** 5
**Confidence:** 4

**Summary:**

The paper presents a feed-forward method for generating clothing-disentangled 3D characters from a single image, reducing the process from hours to seconds. It introduces a two-stage disentanglement approach and a multi-part attention mechanism for high-quality component separation and combination. The authors also contribute a large dataset of over 10k anime characters for training and evaluation.

**Strengths:**

Efficient generation of 3D characters from a single image.

Two-stage disentanglement method for better component separation.

Multi-part attention mechanism for improved information exchange.

**Weaknesses:**

1.	Existing methods, such as ICON[A], HiLo[B], and D-IF[C], do not necessarily require an optimization process when reconstructing a clothed human. What is the superiority of the proposed feed-forward strategy over these methods?
2.	More baseline methods, such as ICON[A], HiLo[B], and D-IF[C], should be considered to fully demonstrate the effectiveness of the proposed method.
3.	In line 303, is obtaining the rotation matrix RcRc dependent on an optimization process?
4.	In Table I, the comparison methods are too few. Moreover, more metrics like clip score, FID (Fréchet Inception Distance), or user studies should be introduced to evaluate the proposed method.
5.	Is it possible to try an input image with a higher resolution, at least one with a clear face?

**Reference**

[A] Xiu, Yuliang, et al. "Icon: Implicit clothed humans obtained from normals." 2022 IEEE/CVF Conference on Computer Vision and Pattern Recognition (CVPR). IEEE, 2022.

[B] Yang, Yifan, et al. "HiLo: Detailed and Robust 3D Clothed Human Reconstruction with High-and Low-Frequency Information of Parametric Models." Proceedings of the IEEE/CVF Conference on Computer Vision and Pattern Recognition. 2024.

[C] Yang, Xueting, et al. "D-if: Uncertainty-aware human digitization via implicit distribution field." Proceedings of the IEEE/CVF International Conference on Computer Vision. 2023.

**Questions:**

Please refer to the weakness part.

---

> ### Author Response · Authors · 2024-11-20
>
> ### Differences Between Our Method and ICON [A], HiLo [B], and D-IF [C]
>
> 1. **Focus on Clothing-Disentangled Generation**
>
>     It is important to clarify that the primary focus of our work is on **clothing-disentangled generation**, rather than treating clothing and the human body as a unified entity, as in [A], [B], and [C]. Our approach addresses this new task by generating not only the complete 3D clothed human model but also the individual 3D models of each clothing item from a single input image. This capability facilitates efficient clothing editing and provides a significant advancement in flexibility and functionality.
>
> 2. **Feed-Forward Generation Without Optimization**
>
>     As demonstrated in Figures 2 and 5 (single GS), our method is fully capable of producing  3D models of both clothed humans and their disentangled components in a **feed-forward manner, without requiring an optimization process.** Optimization, while optional, is introduced in our method as a refinement step to align the individual 3D parts in a unified space, thereby further enhancing rendering quality. This hybrid strategy ensures our approach maintains both efficiency and precision.
>
> ### Baseline Methods
>
> 1. **Clothing-Disentangled Generation as a New Task**
>
>     As mentioned earlier, the task of **clothing-disentangled generation** from a single image is a new problem, and most existing methods are designed to generate only a holistic clothed human. Based on occupancy networks, some works, such as ICON [A], HiLo [B], and D-IF [C], directly predict the holistic clothed human mesh. However, due to the non-watertight nature of VRoidHub 3D models and the complexity of multi-layer occupancy representation, it remains unclear how these methods could be adapted to the task of clothing-disentangled generation.
>
>     In contrast, **multi-view diffusion models** provide a more natural extension to this setting. To this end, we extended the state-of-the-art multi-view diffusion method to serve as our baseline, enabling it to handle multi-part decoupling and generation effectively.
>
> ### Obtaining the Rotation Matrix Rc​
>
> Yes, the optional optimization process is utilized to determine the mutual positional relationships between different parts (Rc, Tc, sc). As illustrated in Figure 5 (right), while the feed-forward strategy is already capable of generating a plausible 3D clothed human model (Single GS), the optimization step further enhances rendering quality (Combined GS).
>
> ### Table 1 Results
>
> Regarding the comparison methods, please refer to the discussion in **Baseline Methods**. As for the evaluation metrics, we follow the conventions established by previous multi-view diffusion studies [Wonder3d]. Thank you for the reviewer’s valuable suggestion. We will incorporate additional metrics in the revised manuscript.
>
>
> ### Image Resolution
>
> Consistent with previous multi-view diffusion methods, we adopt a resolution of 256×256 for generation, primarily due to the limitations of GPU computational resources. When more computational resources become available, higher-resolution inputs can be explored to further enhance the results.

---

> > ### Comment · Reviewer_HfJ1 · 2024-12-03
> >
> > Thank you for your response.  However, I noticed that some of my questions were not addressed directly.  This leads me to believe that the evaluation of the proposed method is still not comprehensive enough. For example, there is no comparison with existing methods using Clip Score. In addition, although ICON, D-IF, and HiLO cannot reconstruct cloth-disentangled 3D humans, it should be feasible to compare the performance with the proposed method using PSNR and SSIM.
> >
> > Based on these concerns, I tend to keep my score.
> >
> > Best regards.

---

> ### Author Response · Authors · 2024-12-03
>
> 1. In line with previous methods [A][B][C] that generate multi-view images from a single image, we adopt PSNR, SSIM, and LPIPS as evaluation metrics to assess image quality. We believe these metrics are sufficient to evaluate the results of our method and the baseline. Regarding the Clip score, it is typically used to evaluate the alignment between images and text. Since our paper focuses solely on image-based input rather than text-based input, we do not find the Clip score to be an appropriate metric in this context.
>
> 2. It is important to emphasize that the main focus of our work is on clothing-disentangled generation, rather than comparing with unified reconstruction methods. As the reviewer has noted, methods such as ICON, D-IF, and HiLO are unable to perform clothing-disentangled generation. Therefore, we do not believe it is necessary to compare our method with these approaches, as it does not detract from the contributions or experimental validation of our paper.
>
>
> Overall, we believe that the experiments and evaluations presented in the paper are sufficiently comprehensive to demonstrate the effectiveness of our proposed method for the task of clothing-disentangled generation.
>
> [A] Wonder3D: Single Image to 3D using Cross-Domain Diffusion
>
> [B] SyncDreamer: Generating Multiview-consistent Images from a Single-view Image
>
> [C] Zero-1-to-3: Zero-shot One Image to 3D Object

---

### Meta-Review · Area_Chair_t9du · 2024-12-18

**Metareview:**

This paper presents a method for generating a 3D character model with disentangled cloth information from a single image. The proposed framework consists of two diffusion models. The first model generates disentangled parts of the 2D image from the given input character image. The second model then generates multiview images of the character conditioned on the disentangled parts. The final 3D model is reconstructed using LGM, a multiview-to-3D generative model, followed by post-optimization. The experiments demonstrate the superior performance of the proposed method compared to Wonder3D, as well as results from an ablation study.

Most reviewers raised concerns about the lack of technical contribution. The only novel technical idea seems to be the two-stage generation process, leveraging disentangled part images as an intermediate representation. The other components largely build on existing generative models and/or fine-tune them. The representation is also specific to a very narrow domain, 3D character generation. Hence, its impact on the general machine learning community would be very limited. As a result, the AC recommends rejection.

**Additional Comments On Reviewer Discussion:**

In the discussion of ACs and SACs, we also suggested the following additions to improve the paper:
-  What enables such disentanglement?
-  How strong is its generalization ability?
-  Single-view 3D reconstruction typically relies on pre-trained models trained on large-scale datasets. What information from these pre-trained models contributes to the results presented in the paper?

---

### Decision · Program_Chairs · 2025-01-22

Reject